# Variations in Rainbow Trout Immune Responses against *A. salmonicida*: Evidence of an Internal Seasonal Clock in *Oncorhynchus mykiss*

**DOI:** 10.3390/biology11020174

**Published:** 2022-01-21

**Authors:** Ruth Montero, Justin Tze Ho Chan, Claudia Müller, Philip Niclas Just, Sven Ostermann, Margareth Øverland, Kevin Maisey, Tomáš Korytář, Bernd Köllner

**Affiliations:** 1Laboratory for Comparative Immunology, Friedrich-Loeffler-Institut, Federal Research Institute for Animal Health, Institute of Immunology, Südufer 10, 17493 Greifswald-Insel Riems, Germany; sven.ostermann@web.de; 2Institute of Parasitology, Biology Centre of the Czech Academy of Sciences, 37005 Czech Budejovice, Czech Republic; justin.chan@paru.cas.cz (J.T.H.C.); tkorytar@paru.cas.cz (T.K.); 3Department of Experimental Animal Facilities and Biorisk Management, Friedrich-Loeffler-Institut, Federal Research Institute for Animal Health, Institute of Immunology, Südufer 10, 17493 Greifswald-Insel Riems, Germany; claudia.mueller@fli.de; 4Aquaculture Research, Bioeconomy, Alfred-Wegener-Institute, Helmholtz Centre for Polar and Marine Research, 27570 Bremerhaven, Germany; Pjust@awi.de; 5Faculty of Biosciences, Department of Animal and Aquacultural Sciences, Norwegian University of Life Sciences, N-1430 Ås, Norway; margareth.overland@nmbu.no; 6Laboratorio de Inmunología Comparativa, Centro de Biotecnología Acuícola (CBA), Universidad de Santiago de Chile, Avenida Libertador Bernardo O’Higgins 3363, Estación Central, Santiago 9170022, Chile; kevin.maisey@usach.cl; 7Faculty of Fisheries and Protection of Waters, University of South Bohemia, 37005 Czech Budejovice, Czech Republic

**Keywords:** seasonality, immune responses, trout, fish inner rhythms

## Abstract

**Simple Summary:**

Our bodies run on an internal schedule or clock, telling us when to rest, sleep, or digest, and when to wake up, be active, or burn calories. That’s why we experience jetlag because we may well set our watches forward or backward, but our bodies haven’t yet. Imagine a seasonal clock that helps get us through the year, not just through the day. We set out to prove that such a clock exists in fish just like it does in humans. We exposed rainbow trout to bacteria to imitate natural encounters. We raised fish in the laboratory under the same light and temperature all year long. When we tested them in summer and winter, the fish consequently experienced days that were artificially longer/shorter or warmer/colder. Nonetheless, certain fish white blood cells didn’t react or see the bacteria as a threat in winter unlike in summer. They were probably behaving based on the time of year, or season and not on their immediate environment, just like how a jetlagged individual behaves based on an internal clock, not on what it’s like outside. Immunity and other processes are regulated differently between seasons, making animals less or more vulnerable in summer or winter.

**Abstract:**

In poikilothermic vertebrates, seasonality influences different immunological parameters such as leukocyte numbers, phagocytic activity, and antibody titers. This phenomenon has been described in different teleost species, with immunological parameters peaking during warmer months and decreased levels during winter. In this study, the cellular immune responses of rainbow trout *(Oncorhynchus mykiss)* kept under constant photoperiod and water temperature against intraperitoneally injected *Aeromonas salmonicida* during the summer and winter were investigated. The kinetics of different leukocyte subpopulations from peritoneal cavity, spleen, and head kidney in response to the bacteria was measured by flow cytometry. Furthermore, the kinetics of induced *A. salmonicida*-specific antibodies was evaluated by ELISA. Despite maintaining the photoperiod and water temperature as constant, different cell baselines were detected in all organs analyzed. During the winter months, B- and T-cell responses were decreased, contrary to what was observed during summer months. However, the specific antibody titers were similar between the two seasons. Natural antibodies, however, were greatly increased 12 h post-injection only during the wintertime. Altogether, our results suggest a bias toward innate immune responses and potential lymphoid immunosuppression in the wintertime in trout. These seasonal differences, despite photoperiod and water temperature being kept constant, suggest an internal inter-seasonal or circannual clock controlling the immune system and physiology of this teleost fish.

## 1. Introduction

All species on our planet have co-evolved with their environment, responding to exogenous stimuli and rhythms, and have developed intrinsic clocks that allow them to anticipate periodic changes and respond accordingly [1]. Different species are influenced by several cycles, such as daily and circadian rhythms; tidal, lunar, and semi-lunar cycles; and seasonality (circannual rhythms) [2]. Circadian rhythms are well studied in mammals and have been characterized at transcriptomic levels [3,4]. This rhythmicity influences physiological parameters such as immune cell levels and composition in the bloodstream or in the organs throughout 24 h cycles [5].

*Aeromonas salmonicida* is the causative agent of furunculosis, an infectious disease affecting wild and farmed salmonids. This bacterium has been widely used as a stimulation and infection model in salmonids [6]. Previously, we described that rainbow trout (*Oncorhynchus mykiss*) exhibit daily diurnal fluctuations in their immune cell composition and respond differently when injected intraperitoneally with *A. salmonicida* at different times of the day [7]. We hypothesize that the immune response against a pathogen could be influenced not only by the time of day but by the time of the year or seasons, as fish also possess circannual rhythms.

Immune functions and the influence of seasonality have been studied in a diverse range of species, including rodents, birds, reptiles [8], and humans [9]. Different approaches have been used, including cell counting between seasons, measurements of organ size and hematological parameters, determination of lysozyme and phagocytic activities, and antibody titers [8]. Recently, researchers have profiled the transcriptomes of human blood samples [10] and, interestingly enough, of the teleost fish three-spined stickleback (*Gasterosteus aculeatus*) [11,12]. Brown et al. [11] showed a strong circannual oscillation of immune-related genes in the stickleback, measuring the expression of sets of genes that are winter- or summer-biased. Among the summer-biased transcripts, they found genes related to adaptive immune responses (*rag1*, *rag2*, *zap-70*, *cd8 a*, *foxp3b*, *il4*, *igh*), while the winter-skewed ones had higher levels of innate immune (*il1* and non-classical complement pathways) and lymphocyte immunosuppressive genes.

Specifically in rainbow trout, leukocyte numbers peak during summertime and are lower in the winter [13,14,15]. Morgan et al. broadened the seasonal evaluation of parameters and measured, for 12 months, the white and red blood cells counts, plasma lysozyme activity, and respiratory burst of head kidney cells [16]. Erythrocytes showed a peak in warm months (June and July), and total white blood cells were significantly higher in the summer months than in the winter. Plasma lysozyme activity also followed this pattern, and no clear seasonal trend was observed for respiratory burst activity. Considering this, it is hypothesized that in general, all parameters are diminished in winter and increased in summer [17]. Water temperature and photoperiod are key factors that influence the seasonal variation of immune parameters [14]. However, some studies have shown changes in immune parameters even when temperature and photoperiod were constant, suggesting that endogenous rhythms may also be involved [18]. Despite this, to our knowledge, there are no studies evaluating differences in the composition of leukocyte subsets in warm or cold seasons, nor studies examining the capacity to respond to the antigenic challenge among seasons, nor studies investigating if the inner clocks of fish regulate their immune responses circannually while maintaining the same water temperature and photoperiod.

In this study, we answer some of these questions by stimulating rainbow trout intraperitoneally (i.p.) with inactivated *A. salmonicida* while keeping water temperature and photoperiod constant over the winter and summer months. We measured how immune cell populations responded to the bacterial stimulus by flow cytometry, and additionally, we evaluated the antibody response in both conditions.

## 2. Materials and Methods

### 2.1. Ethics Statement

The experiment was approved by the State Office for Agriculture, Food Safety and Fisheries (approval number LALLF 7221.3-2-042/17), according to the German and European guidelines on animal welfare (Tierschutzgesetz, Tierschutz-Versuchstierverordnung, Directive 2010/63/EU).

### 2.2. A. salmonicida for Stimulation Experiments

An aliquot of the *A. salmonicida ssp. salmonicida* highly virulent strain JF 5505 from stock cryo-preserved batches—previously checked for purity by Gram staining, cell morphology, and motility—was cultivated in tryptic soy broth media (TSB, Becton Dickinson, Heidelberg, Germany) at 15 °C for 24 h. The bacterial suspension was inactivated in 1.5% paraformaldehyde (PFA) for 1.5 h at 4 °C. Inactivated bacteria were plated out on TSB agar plates to verify that no bacteria grew after inactivation. The PFA was washed away twice with TSB media, each time centrifuging at 4000 *g* for 10 min at 4 °C. The pellet was resuspended in TSB 25% glycerol at a concentration of 1.5 × 10^8^ bacteria/mL. For intraperitoneal immunization, the bacteria were washed once with PBS and set to a concentration of 1 × 10^7^ bacteria/mL. Injections were prepared under aseptic conditions with sterile 1× phosphate-buffered saline (PBS).

### 2.3. Fish

The Born strain of rainbow trout (*O. mykiss)* was bred and purchased in Germany, with no gender selection, from the commercial trout breeding farm Forellenzucht Uthoff GmbH, Neubrandenburg (Germany). The fish were kept in 300 L glass aquaria in a partially recirculating water system, at constant 12 ± 0.2 °C and constant 12 h light: 12 h dark period for both summer and winter experiments. Dissolved oxygen (11 ± 0.5) and pH (7.1 ± 0.1) were monitored daily. Ammonia levels never surpassed 0.1 mg/L. Summer experiments were performed in September; winter experiments were performed in January, European time zone. Fish were fed twice per day with commercial dry food pellets (Aller Futura, Aller Aqua GmbH Golssen, Germany). A total of 130 fish randomly distributed through the groups were used for the experiments, without special criteria for bias. Fish weighed 26.4 ± 12.7 g at the beginning of the trial and weighed 36.4 ± 21.3 g at the end of the trial (group 28 days post-i.p. injection). Five fish were used for the cell baselines in summer and five fish for the cell baselines in winter (ten in total). A total of 120 fish were used for sampling at the different time points for both control PBS-injected fish and for the bacteria-injected fish.

The fish were separated according to the sampling time points and received a single i.p. injection, containing 100 µL of 1 × 10^7^ bacteria/mL or 100 µL of PBS (control fish). After i.p. injection, control and stimulated fish were kept in separated tanks. The tanks were divided into two areas (one per time point) using a plastic mesh, in order to keep fish separated. Five fish were used at each time point per condition (12 h, 24 h, 48 h, 72 h, 14 d, and 28 d post-i.p. stimulation with inactivated bacteria or injection with PBS) for a total of 60 per season.

### 2.4. Sampling and Leukocyte Preparation

On the day of sampling, fish were sacrificed with an overdose of benzocaine (Sigma, Gernsheim, Germany). Blood was taken from the caudal vein, collected in EDTA-containing tubes (Sarstedt AF & Co., Nümbrecht, Germany) and kept on ice until processing. Thymus, spleen, head kidney, and peritoneal cells were collected. All cell processing was performed on ice and all reagents were ice-cold. Blood volume, peritoneal lavage volume, and fish and organ weights were measured and recorded for each animal.

Peritoneal leukocytes (PELs) were obtained via lavage with 5 mL of ice-cold 5 mM EDTA-PBS. Leukocytes from the other organs were obtained after homogenization in 5 mL of 1% newborn calf serum (NCS)-PBS buffer (FACS Buffer, FB). Blood was washed with 5 mL of FB. All cell suspensions were centrifuged at 4 °C for 5 min at 290 *g* and then resuspended in 1 mL of FB. A hypotonic lysis protocol for erythrocytes was modified from Crippen et al. [19] and optimized for each organ sampled. Briefly, erythrocytes were disrupted by adding 8 mL of ice-cold Milli-Q water and mixing by inversion for 5 s for the PELs and 10 s for spleen and head kidney; 1 mL of 10× PBS was added to restore the isotonicity. The cell suspension was placed on ice for 5 min, allowing debris to clump and precipitate. Next, the leukocytes were filtered through a 100 μm mesh and then centrifuged. The pellet was finally resuspended in 1 mL of FB and then the living cells were counted by trypan blue exclusion (Gibco, Thermo Fisher, Bremen, Germany).

The cell composition of the above-mentioned tissues was analyzed using flow cytometry. Additionally, the kinetics of the *A. salmonicida*-specific antibodies were measured by Enzyme-Linked Immunosorbent Assay (ELISA).

### 2.5. Flow Cytometry

A multicolor flow cytometry approach was used to analyze the immune cell kinetics and distribution in the different tissues. All monoclonal antibodies were previously validated [20]. Briefly, a first blocking step was made prior to antibody staining, incubating 4 × 10^5^ cells per tube with FB for 30 min at 4 °C. The cells were then pelleted by centrifugation at 290× *g* for 5 min at 4 °C, and the pellet was resuspended in 200 µL of antibody solution containing anti-pan T-cell monoclonal antibody D30 (mAb D30) and anti-trout CD8α. Cells were incubated for 30 min at 4 °C and then washed with 700 µL of FB by centrifugation as described above. The cell pellet was resuspended in 200 µL of secondary antibody solution (anti-mouse IgG1 405 (Jackson Immuno Research, Biozol, Eching, Germany; anti-rat IgG-Alexa 647 (Invitrogen, Fisher Scientific GmbH, Schwerte, Germany)). After an incubation of 30 min at 4 °C, a washing step was performed as described. The cells were resuspended in 200 µL of a mixture of directly labeled monoclonal antibodies: anti-Igµ chain monoclonal antibody 1.14 (mAb 1.14), anti-light chain monoclonal antibody N2 (mAb N2), anti-myeloid cell monoclonal antibody 21 (mAb 21), and anti-thrombocyte monoclonal antibody 42 (mAb 42, to exclude this population from the analysis). Cells were incubated for 30 min at 4 °C, washed, and finally incubated with the viability dye Zombie Aqua (BioLegend, San Diego, CA, USA) at a dilution of 1:1000 in 1× PBS for 20 min at 4 °C. After a final wash step, the cell pellet was resuspended in 300 µL of FB. Cells were analyzed by the Cell Analyzer BD LSR Fortessa (Becton Dickinson, Germany), recording a minimum of 30,000 events for each sample. Doublet discrimination was performed in dot plots using the parameters FSC-H vs. FSC-A (forward scatter height versus forward scatter area) and SSC-H vs. SSC-A (side scatter height versus side scatter area). Lymphoid (FSC^low^ SSC^low^) and myeloid cells (FSC^hi^ SSC^hi^) were distinguished by their scatter profiles. The cytometric analysis was made in the BD FACSDIVA software (BD Biosciences, Heidelberg, Germany).

### 2.6. ELISA

To measure specific anti-*A. salmonicida* IgM antibodies, high-binding ELISA plates (Greiner, Sigma, Germany) were coated with 1 µg per well of inactivated *A. salmonicida* in 0.2 M Carbonate buffer pH 9.6 overnight at room temperature (RT) and then blocked with 150 µL of ROTI^®^Block solution (Carl Roth, Karlsruhe, Germany) for 1.5 h at RT. A total of 100 µL of a two-fold dilution series of 1:10 pre-diluted fish sera from 1:80 to 1:81,920 was incubated for 1 h at RT. After three washing steps with PBS 0.05% Tween 20 (PBST), 200 ng per well of purified anti-trout IgM antibody 4C10 was incubated for 1 h at RT, followed by three washing steps with PBST. Finally, the secondary antibody goat anti-mouse IgG1 HRP conjugated 1:5000 (mouse IgG2a, IgG2b, IgG3, IgM, IgA; human, bovine, and horse serum proteins cross-adsorbed, Southern Biotech, Biozol, Eching, Germany) was incubated for 1 h at RT. After three washing steps with PBST, the wells were developed by a 10-min incubation at RT with 50 µL of TMB/E Horseradish Peroxidase Substrate (Merck, Darmstadt, Germany). Finally, an additional 50 µL of H_2_SO_4_ was added to stop the reaction, and the optical density of each well was measured at 450 nm in the Tecan ELISA plate reader (Tecan Sales Austria GmbH, Grödig, Austria). To measure “natural antibodies”, plates were coated with 100 ng of 2,4-Dinitrophenyl-Keyhole Limpet Hemocyanin (DNP-KLH, Merck, Darmstadt, Germany). The ELISA protocol used was identical to the one above.

## 3. Results

### 3.1. Seasonal Cell Composition of Unstimulated Fish

Figure 1a shows the stepwise gating strategy we used to define the lymphocyte and myeloid gates. Figure 1b shows representative morphometric plots (forward scatter versus side scatter, or size versus intracellular complexity in other words) of leukocytes in each organ analyzed as well as the respective lymphoid and myeloid gates that we defined. Analyzing the number of leukocytes in the peritoneum (Figure 1c), significantly higher numbers of myeloid cells were detected in the wintertime, whereas no statistically significant difference was measured for the lymphoid population. In the spleen, head kidney, and thymus, significantly higher numbers of lymphocytes were detected during the summer season. Among these tissues, we measured a decrease in the number of myeloid cells only in the thymus (*p* > 0.05). Thus, under identical experimental photoperiod and water temperature, we nonetheless measured inter-seasonal variation in lymphoid cell numbers in all lymphoid organs tested as well as in myeloid cell numbers in the peritoneum and thymus.

### 3.2. Seasonal Cell Composition of the Peritoneal Cavities of A. salmonicida-Stimulated Fish

The differences in the cell numbers in the different lymphoid organs between the two seasons led us to test if this influences the cellular kinetics in response to a pathogen. For this purpose, we stimulated fish i.p. with fixed *A. salmonicida* strain JF5505 and we followed up the cell kinetics locally (peritoneal cavity) and systemically (spleen and head kidney) after 12 h, 24 h, 48 h, 72 h, 14 d, and 28 d post-stimulation.

In the peritoneal cavity in the summer (Figure 2a, upper panels), during the first 24 h post-stimulation (hps) with the bacteria, the intraperitoneal stimulation induced a clear switch in the predominant local cell type (Figure 2a); the myeloid population increased starting at 12 hps and peaked at 24 hps. As the myeloid population started to drop down 48 hps, the lymphoid population started increasing at this time point, reaching a peak at 72 hps. In the late response phase, i.e., 14 and 28 days post-stimulation (dps), the cell levels in the peritoneal cavity returned to baseline and were comparable to those of the PBS-injected group. In the winter (Figure 2a, lower panels), we instead observed a simultaneous increase in both myeloid and lymphoid cells, starting at 12 hps and reaching a peak at 48 hps. Afterwards, the myeloid fraction dropped down faster than the lymphoid one, which remained slightly higher at 72 hps, but not statistically significantly. As in the summer, at 14 and 28 dps, both levels were back to baseline.

### 3.3. Seasonal Cell Composition in the Spleens and Head Kidneys of A. salmonicida-Stimulated Fish

In the spleen during the summer (Figure 2b, upper panels), no statistically significant variation was observed in the myeloid cell population. However, a significant increase in the lymphoid population was observed in the spleen at a single time point (24 hps) before returning to baseline levels. In the wintertime (Figure 2b, bottom panels), no response was observed for the lymphoid nor myeloid cells at any time point.

In the head kidney (Figure 2c, upper panels), the lymphoid cell numbers increased significantly only at a single time point, as in the spleen, except it was very much delayed to 14 dps. Just like in the spleen, no significant changes were measured for the myeloid population (Figure 2c, top panel) nor any changes in these populations during the wintertime (Figure 2c, bottom panels).

### 3.4. Response of Leukocyte Subpopulations in the Peritoneal Cavities of Stimulated Fish

To determine which subpopulation was responsible for the observed changes in the response of lymphoid and myeloid cells, the leukocytes were labeled with different leukocyte population-specific antibodies and analyzed by flow cytometry.

The response of B-cell populations in the peritoneal cavity to *A. salmonicida* stimulation was different between the summer and winter. In the summer, an increase in the IgM^−^ (IgT^+^) B-cell population between 24 h and 72 h post-injection was observed, and the same occurred with the IgM^+^ lymphocytes (Figure 3a,b, left column). In comparison, in the wintertime, we observed a peak in IgM^+^ lymphocytes only at 48 h post-injection, whereas no IgT response was observed (Figure 3a,b, right column).

For the CD8^+^ T-cell population, delayed responses were measured at 72 hps in the summer and maintained on day 14 post-stimulation, returning to baseline levels thereafter (Figure 3c). In contrast, no significant change was observed in winter for the CD8^+^ T cells nor among CD8^−^ T cells (that are likely CD4^+^ T cells) at any time point or in any season (data not shown).

During the summer, the local response of the myeloid population (Figure 3d, left) showed an early increase in the cell numbers at 12 hps, reaching a peak in cellularity at 24 hps. At 48 hps, the response already began decreasing while remaining higher than the baseline and remained so until 72 hps. In winter, the response observed in the first 72 h was stronger than the one observed in summer (Figure 3d right). A higher early increase in cell numbers at 12 hps was observed and increased gradually, peaking at 48 hps. Thus, in response to *A. salmonicida* stimulation, the myeloid compartment expanded and peaked rapidly in the summer, decreasing gradually, whereas the change in the season led to a gradual expansion and peak, with a sudden drop at 72 hps.

Our results matched those presented in Figure 2 for the peritoneal cavity, with cellular kinetics suggesting the phenomenon of an initial myeloid phase response during the first 24 h in the summer, followed by a delayed lymphoid phase response. In winter, only a synchronized cell increase between myeloid and lymphoid cells (predominantly IgM^+^ lymphocytes) was observed, peaking at 48 hps.

### 3.5. Response of Leukocyte Subpopulations in the Spleens of Stimulated Fish

Shifting our attention to the spleen after fish were stimulated with *A. salmonicida*, a rapid, significant, and short-lived increase in the number of cells per µg of tissue was observed in the summer at 24 hps in both B-cell populations (Figure 4a,b, left column) relative to PBS-injected individuals. On the contrary, no trend or statistically significant change was observed during the wintertime (Figure 4a,b, right column).

Myeloid cells (labeled with mAb 21) increased at 24 hps during the summer (Figure 4c, left panel) and mirrored the lymphocyte compartment with this change or any change being completely absent in winter.

### 3.6. Response of Leukocyte Subpopulations in the Head Kidneys of Stimulated Fish

In the head kidney, there were variable increases in the number of IgT^+^ and IgM^+^ lymphocytes during the summer at 12 and 24 hps and at 14 and 28 dps (Figure 5a,b, left column). Meanwhile, in the winter, no statistically significant responses were measured at any time point for either of these compartments. (Figure 5a,b, right column).

On the other hand, higher cell numbers of CD8^−^ T cells (CD4^+^ T cells) were observed at 28 dps only during summer (Figure 5c, left); no trend nor difference between PBS- and *A. salmonicida*-injected fish were noted during winter within this cell population (Figure 5c, right panel).

Myeloid cells showed no change at any time point regardless of the season or the stimulus (Figure 5d).

### 3.7. Specific and Non-Specific Antibody Titers in Fish Stimulated with A. salmonicida

Due to the significant changes in the B-cell compartment and the expected contribution of B cells to the response against the bacteria, we evaluated specific and non-specific antibody production by ELISA across all the time points used.

During the summertime (Figure 6, left graph), minimal non-specific antibodies were detected during the earliest time points evaluated (12–72 hps). A slight increase was observed at 14 dps (titer 1:160) and 28 dps; the titer increased to 1:320. With regard to the specific antibodies, they were not detected until day 14 post-stimulation, with a titer of 1:640. The titer remained at 1:640 by day 28 post-stimulation.

During the wintertime (Figure 6, graph on the right) at 12 hps, an unusually high titer of non-specific antibodies was present in the serum of fish injected with the bacteria (1:5120). After 24 h, the titer rapidly decreased to 1:320 and remained so until 48 hps. Only in the wintertime during the first 12 h after the injection with the bacteria was this exponentially larger non-specific response observed. The specific antibodies were detected at 14 dps with a titer of 1:640, remaining in that range until 28 dps, identical to what we observed in the summertime.

## 4. Discussion

As animals do not control their environment, they develop strategies to anticipate and cope with seasonal shifts. This way, they are prepared to respond accordingly at the molecular, cellular, and metabolic levels, favoring survival and maintaining organism fitness [21]. We observed interseasonal differences in how fish responded or did not respond to stimuli. Although these experiments need to be repeated and expanded upon with more fish and more photoperiod/temperature conditions, it nonetheless warrants discussion and speculation as to how our findings compare to the very limited number of publications on the effect of seasonality on fish immunity, what specific cell subsets are most affected between seasons, and the mechanisms behind a potential circannual clock.

### 4.1. Seasonal Immune Status: Immune Parameters Influenced by the Season

A study in the three-spined stickleback (*Gasterosteus aculeatus*) reported that immune markers were winter- or summer-biased, observing suppression of adaptive immune and lymphocyte proliferation genes during winter [11]. In *O. mykiss*, changes have been reported in the baselines of immune cells, lysozyme and phagocytic activities, and antibody titers between winter and summer [16]. Additionally, it was reported that erythrocyte and leukocyte numbers peak during summer and are the lowest during winter [13,14,15].

Our initial analysis of the cell baselines showed that during summer, spleen, head kidney, and thymus had more lymphocytes than in winter. This suggests that lymphocytes are actively being produced during this season and are home to organs such as the spleen and head kidney. Fish lack lymph nodes; therefore, these two organs are probably where the T-cell and antigen-presenting cell (APC) interactions happen [22]. Interestingly, in the peritoneum, lymphoid cell numbers were similar during both seasons; however, the myeloid cell population was increased during the wintertime. The decreased number of lymphoid cells in the winter suggests reduced hematopoietic activities in the thymus and head kidney and potentially a limited proliferative activity in the spleen. Fish thymuses involute during winter months because of their season-dependent activity, meaning thymus thickness decreases, has low lymphocyte numbers, and has more adipose tissue and collagenous fibers [23,24,25,26]. Considering this, we would expect less T lymphocytes circulating during these months, contributing to a long-term reduced immune response activity in poikilothermic fish. Myeloid cells are rapid and first responders limiting the spread of invading pathogens. The higher number of myeloid cells in the peritoneal cavity might be an adaption to wintertime. Perhaps resident myeloid cells are preferred over constantly recirculating cells for this season.

### 4.2. Peritoneal Cavity: Biphasic Response in Summer, Joint Response in Winter

From the peritoneal cavity, the development of a biphasic leukocyte response was clear after i.p. stimulation: a switch from a myeloid- to a lymphoid-dominant phase at 48 hps during summer. These leukocyte kinetics correspond with what has already been published by others using a stimulation and infection model with *A. salmonicida* [27]. Thus, we hypothesize that during summer, the lymphocytes residing in the peritoneal cavity contribute to the immune response upon the bacterial stimulus. They secrete proinflammatory cytokines that, within the first 24 h, help to recruit myeloid cells to the cavity [28,29]. In winter, a different scenario was evident. A higher simultaneous increase in lymphoid and myeloid cells was observed until 48 hps. In this situation, pathogen detection and cytokine release by the two populations were probably concomitant and increased in the winter. Consequently, much higher cell numbers/mL of peritoneal wash were reached, suggesting a stronger inflammatory process occurred from which fish should recover afterwards.

### 4.3. Immune Response Skewed toward Innate Immune Responses in Winter in Trout

T and B lymphocytes are described as the main resident populations in the peritoneum of mice; one resident B cell subpopulation is B1 B cells [30]. They sense pathogen-associated molecular patterns (PAMPs) through TLRs, act as APCs and phagocytes, secrete cytokines for cell recruitment, and secrete high amounts of low-affinity poly-reactive antibodies [31,32]. Even though B1 B cells are not fully described in fish, the first hints of their presence in zebrafish [33] and rainbow trout [34] were already reported. Considering the immune functions of B1 B cells, it would be reasonable to hypothesize that B1 B cells reside in the peritoneal cavity of fish.

From our results, we can observe a strong peritoneal IgM^+^ response in winter, maybe due to the presence of a higher number of B1 B-like cells during this season, since this subset expresses high IgM on their membranes [35]. As these cells are innate-like and we also observed more myeloid cells present during winter, we hypothesize that during the winter there is a shift toward an innate status of this cavity to combat pathogens. This could be due to the need to manage metabolism and the trade-offs needed to balance investment in immunity versus other physiological processes to promote survival [36]. Colder months are already challenging for fish, with limited nutrient and oxygen availability among others, such that fish need to compensate by, for example, limited activity and lower heart rate. From an immunological perspective, these types of survival decisions have been reported in different species, including birds [37] and humans [38]. In mice, it has been shown that the initial generation of plasmablasts against malaria is metabolically taxing, to the point of compromising the generation of memory B cells and favoring the progression of the parasite life cycle [39]. Innate responses require lower initial energetic demand and are not energetically costly to maintain, which is why they are favored in winter months. Additionally, they constitute a fast and effective responder system against new pathogens [38]. Overall, this suggests a regulation of adaptive immunity during the winter, rather than a loss of function.

### 4.4. Seasonal and Circannual Rhythms Modulate the Immune Response of Trout

In lower ectothermic vertebrates, including fish, lower peripheral blood lymphocyte counts during the winter and higher numbers during the summer are general trends, except in *Salmo trutta*, in which the parameters were reversed [18]. In teleosts, three different isotypes—IgM, IgD, and IgT—have been described [40]. In this study, we analyzed IgM^+^ cells and IgM^−^, mostly and likely IgT^+^ cells [41]. We observed a mixed response between these two B-cell subsets only during summer, which suggests that immune fitness and response capacity are higher during the summer months and that the mucosal response mediated by IgT is suppressed in winter. Moreover, only in summer, the head kidney showed an increase in the IgT^+^ population 14 and 28 days after stimulation. Since we lack antibodies against IgT, we could not measure IgT levels in serum to at least have an indication of whether plasma cells could also play a role in this response.

### 4.5. Temperature and Photoperiod: Key Parameters Influencing Immune Seasonal Variations in Trout

On the one hand, it is known that temperature affects B cells and antibody secretion [42]. In this study, water temperature was constant at 12 °C in both seasons, and the titer of specific IgM antibodies against *A. salmonicida* were similar in the two periods. However, interestingly, non-specific antibodies were highly increased 12 h after the bacterial stimulation and gradually decreased afterwards. Together, this also suggests that more B1 B cells could be present during winter in the peritoneal cavity, secreting natural non-specific antibodies to support the immune response against pathogens, since fish are not in an optimal immune state.

On the other hand, we observed that despite maintaining the water temperature and photoperiod constant, different cellular immune responses were observed in the two seasons. This has been reported previously for mice [43,44] and also for lower vertebrates [18,45], suggesting that neuroendocrine and other endogenous rhythms are involved in regulating these responses. It is known that adrenocortical activity is modulated on a circannual basis, having a higher corticosteroid secretion in winter and a lower one during summer, directly affecting immune parameters. Glucocorticoids suppress cellular immune function, affecting T cells to a greater extent. Thus, when glucocorticoids are elevated, T cell activity is suppressed, and B cell activity is elevated [46]. In fish, it was described that cortisol, the primary corticosteroid produced in teleosts, exerts strong anti-inflammatory effects and inhibits pro-inflammatory cytokines [47]. In vitro studies made in gilthead seabream [48], sea bass, and rainbow trout [49] showed that the cortisol effect is immunosuppressive and may enhance disease susceptibility. In rainbow trout, cortisol reduced the number of circulating lymphocytes; decreased lymphocyte proliferation, antibody production, and phagocytosis; and increased apoptosis [50]. This is in line with what we observed during winter, where T cell responses were dramatically decreased. However, the effects of hormones/cortisol are not fully depleting but rather immunosuppressive for B cell lymphopoiesis, sparing cortisol-resistant long-lived memory effector B cell populations. Probably, this effect is to favor the survival of spawning fish upon return to their natal streams. Thus, similarly, the immunosuppressive effects of cortisol in winter may not directly impact the survival of fish if neither their compartment of differentiated, highly selected effector memory cells nor the antibodies they produce are compromised, with only less (excessive) energy expanded on lymphopoiesis [51].

### 4.6. Indirect Evidence of a Potential Internal Circannual Clock in Trout

Marine animals, including teleost fish, are highly regulated by circadian, circalunar, and circannual rhythms [52,53]. Cell-autonomous, long-term timekeeping systems (circannual rhythms) are endogenous calendars that animals have maintained throughout evolution to track physiological and behavioral cycles and to anticipate and sense environmental variations that are crucial for their survival [54,55,56,57]. How these annual clocks work is still unknown. It has been proposed that circannual rhythm generation depends on tissue-autonomous cell regulation, repetitive cell division cycles, functional cell differentiation, and cell death [58]. In rainbow trout, circadian rhythm-related genes such as *Clock1a, Bmal1, and Per1* have been described in the retina and hypothalamus [59]. Thus, as environmental cues are not necessary to modulate this clock, the circannual biological pacemakers run untethered, even if stimuli such as light or temperature are experimentally altered. As we observed in our experimental conditions, even keeping photoperiod and water temperatures constant resulted in different cell baselines and a different immune response against the bacterial pathogen. Further studies to elucidate which transcription factor(s) modulate the circannual modulation of the immune response should be performed and, using these markers, tissue specificity should be investigated.

## 5. Conclusions

Altogether, this work shows a difference in the immune response against a bacterial stimulus between two seasons, despite environmental cues such as water temperature and photoperiod remaining unchanged. It provides information suggesting rainbow trout may have circannual clocks that modulate physiological parameters, including immune cell levels, that may impact immune responses directly. Moreover, we only measured mixed innate and adaptive cell responses during the summer. These important findings should be considered in three main contexts: (a) to update “when” and under which conditions we perform our laboratory experiments in order to improve reproducibility; (b) to optimize “when” we apply stimuli to fish (if we consider vaccination as a stimulus, based on our results, we should vaccinate fish during summer months, coinciding with fish being at their most immunocompetent status); and (c) to improve fish health when they are raised under artificial conditions. Recirculating aquaculture systems frequently use unnatural photoperiod regimens such as 24 h of light and 0 h of darkness (24L:0D) or 16L:8D for extended periods. Although the reasoning behind this is to improve the growth rate, it is thought that fish could also be immunologically protected, as summer conditions are mimicked. However, our results indicate that despite the attempt to “manipulate” the physiology of fish, this may not be possible due to their inner clocks. Finally, relevant topics remain to be investigated further, such as elucidating how different photoperiod regimens and their durations influence the immune status and immune responses of fish, to promote immune protection and decrease susceptibility to pathogens.

## Figures and Tables

**Figure 1 biology-11-00174-f001:**
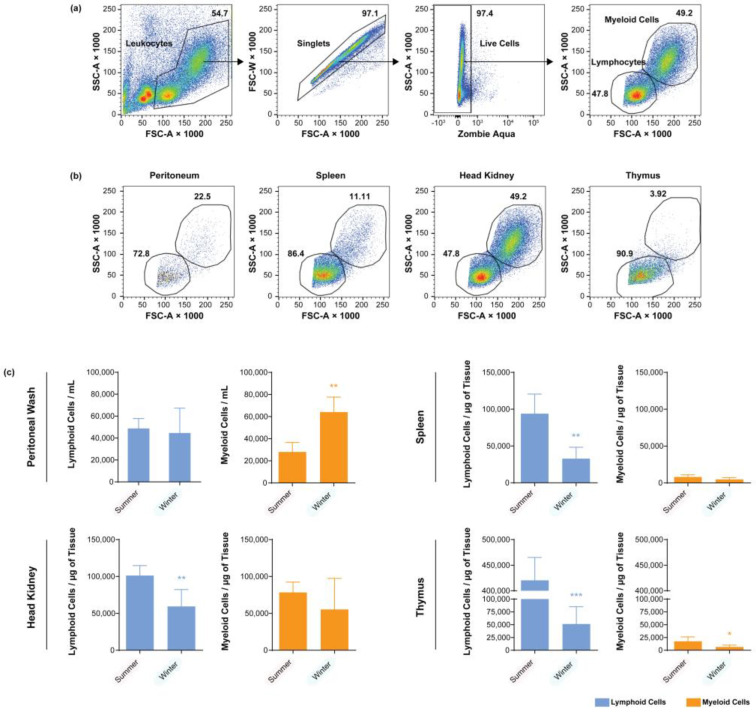
Unstimulated rainbow trout demonstrate different lymphoid and myeloid cell counts between the summer and winter seasons in the peritoneal cavity and lymphoid organs. (**a**) Gating strategy used for the flow cytometric analysis. (**b**) Representative morphometric plots of each organ analyzed as well as the gates established to distinguish lymphoid from myeloid lineage cells. (**c**) Lymphoid and myeloid cell counts. Bars are color-coded to represent either lymphoid (blue) or myeloid (orange) cells. Note the decrease in lymphocytes in spleen, head kidney, and thymus during winter. Any statistical significance was determined with Student’s *t*-tests; *n* = 5 per group per season. * indicates *p* < 0.05; ** indicates *p* < 0.01; *** indicates *p* < 0.001. The data are presented as mean values ± standard deviation (SD).

**Figure 2 biology-11-00174-f002:**
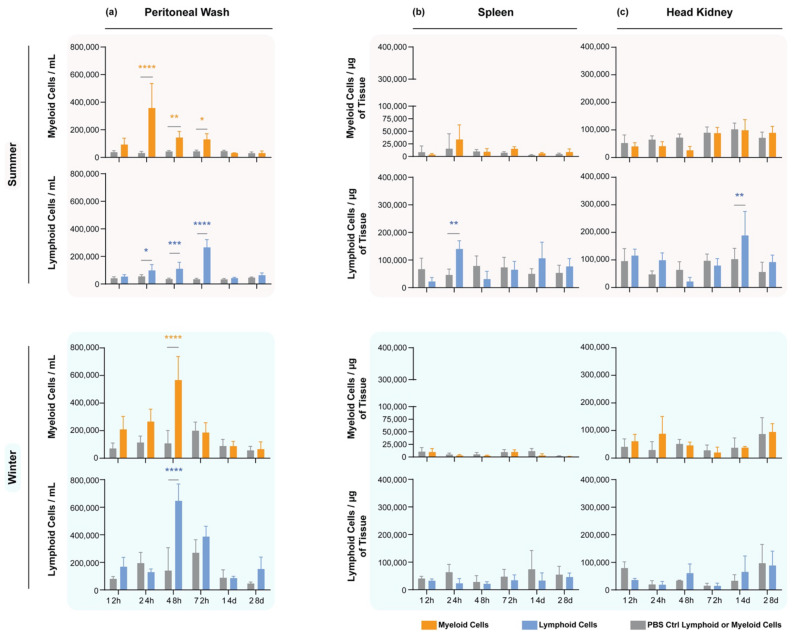
Myeloid and lymphoid cells at different time points after stimulation with *A. salmonicida*. Gray bars represent control fish injected with PBS. (**a**) Peritoneum, (**b**) spleen, (**c**) head kidney stimulated in summer (two upper rows) and winter (two lower rows) under constant water temperature and photoperiod. The local peritoneal response in summer was composed of an orchestrated response, with a myeloid-dominant phase in the first 24 hps, switching to a lymphoid phase at 48 hps. In winter, the peritoneal leukocytes increased, jointly reaching a peak at 48 hps. The spleen responded faster than the head kidney in the summertime; however, in the winter, no response was observed in both organs. Y-axes depict the number of cells per mL of peritoneal wash or the number of cells normalized to the amount of tissue collected (for the lymphoid organs), whereas the x-axes represent the number of days after administration of either *A. salmonicida* or PBS. Bars are color-coded to represent either lymphoid (blue) or myeloid (yellow) cell counts after stimulation, or (gray) their corresponding groups of lymphoid or myeloid cells from fish injected with PBS. Statistical significance was calculated with a two-way ANOVA multiple comparison test, with a Bonferroni post hoc correction test. *n* = 5 for each time point. * indicates *p* < 0.05; ** indicates *p* < 0.01; *** indicates *p* < 0.001; **** indicates *p* < 0.0001. The data are presented as mean values ± standard deviation (SD).

**Figure 3 biology-11-00174-f003:**
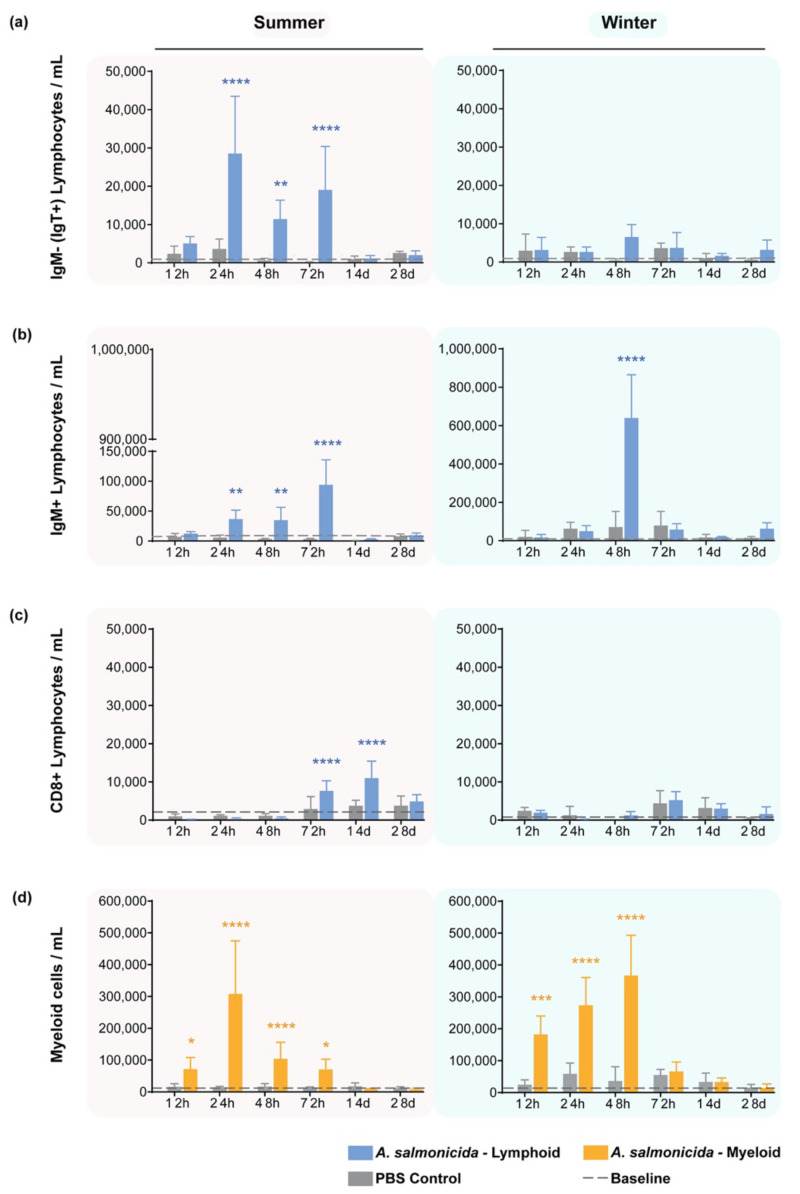
Response of peritoneal cells after stimulation with *A. salmonicida* analyzed by flow cytometry. (**a**) IgM^-^ lymphocytes/mL; (**b**) IgM^+^ lymphocytes/mL; (**c**) CD8^+^ lymphocytes/mL; (**d**) myeloid cells/mL in summer (**left column**) and in winter (**right column**). Whereas IgM^+^ B cells responded in both seasons, the IgM^−^ (IgT^+^) cells increased after stimulation in the summer only. CD8^+^ T cells also responded exclusively in the summer. In the summertime, myeloid cells responded as early as 12 hps, whereas in the winter, the recruitment peaked at 48 hps and dramatically decreased after this time point. Baseline levels of the cells are from 5 naïve fish. Y-axes depict the number of cells per mL of peritoneal wash, whereas the x-axes represent the number of days after administration of either *A. salmonicida* or PBS. Bars are color-coded to represent either lymphoid (blue) or myeloid (yellow) cell numbers from fish stimulated with the bacterium. Gray bars are the corresponding lymphoid or myeloid cell counts from fish injected with PBS. Statistical significance was calculated with a two-way ANOVA multiple comparisons test, with a Bonferroni post hoc correction test. *n* = 5 for each time point. * indicates *p* < 0.05; ** indicates *p* < 0.01; *** indicates *p* < 0.001; **** indicates *p* < 0.0001. The data are presented as mean values ± standard deviation (SD).

**Figure 4 biology-11-00174-f004:**
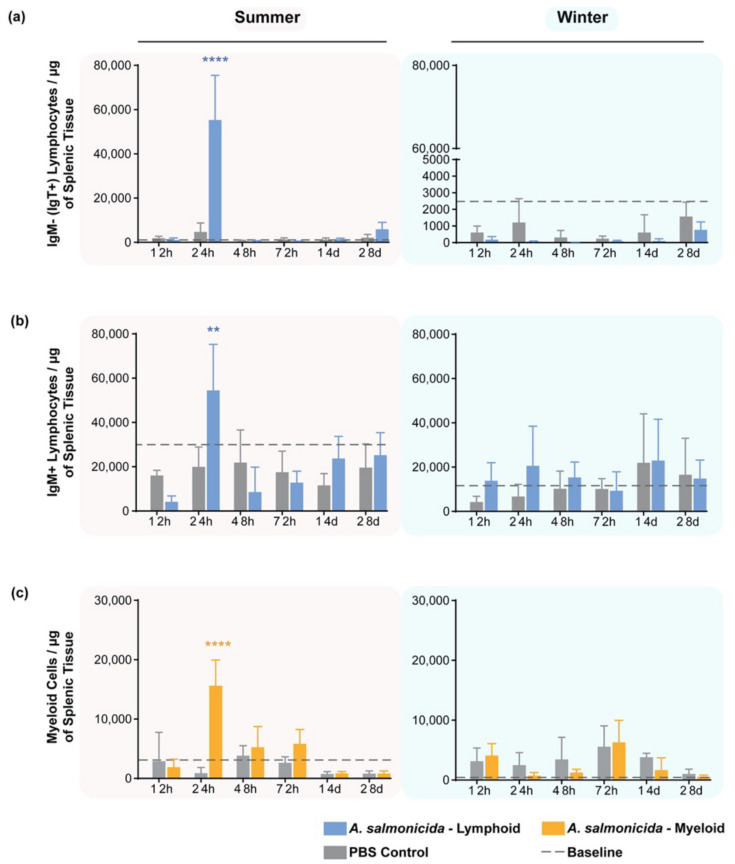
Cell response in the spleen after stimulation with *A. salmonicida* analyzed by flow cytometry. (**a**) IgM^-^ lymphocytes/μg of tissue; (**b**) IgM^+^ lymphocytes/μg of tissue; (**c**) myeloid cells/μg of tissue measured in summer (**left column**) and in winter (**right column**). An increase in the number of Ig^+^ cells was only observed during summer (at 24 hps) for both Ig isotypes. Myeloid cell numbers peaked at 24 hps in summer. No statistically significant peak of response was observed during the wintertime. Baseline levels of the cells are from 5 naïve fish. The bars are colored blue to represent the number of lymphoid cells from *A. salmonicida*-exposed animals or the number of myeloid cells (yellow) from the same animals. Y-axes represent the number of cells per µg of splenic tissue, whereas the x-axes represent the number of days since fish were injected. Gray bars represent the corresponding lymphoid or myeloid cells from groups of fish injected with PBS. Statistical significance was calculated with a two-way ANOVA multiple comparisons test, with a Bonferroni post hoc correction test. *n* = 5 for each time point. ** indicates *p* < 0.01; **** indicates *p* < 0.0001. The data are presented as mean values ± standard deviation (SD).

**Figure 5 biology-11-00174-f005:**
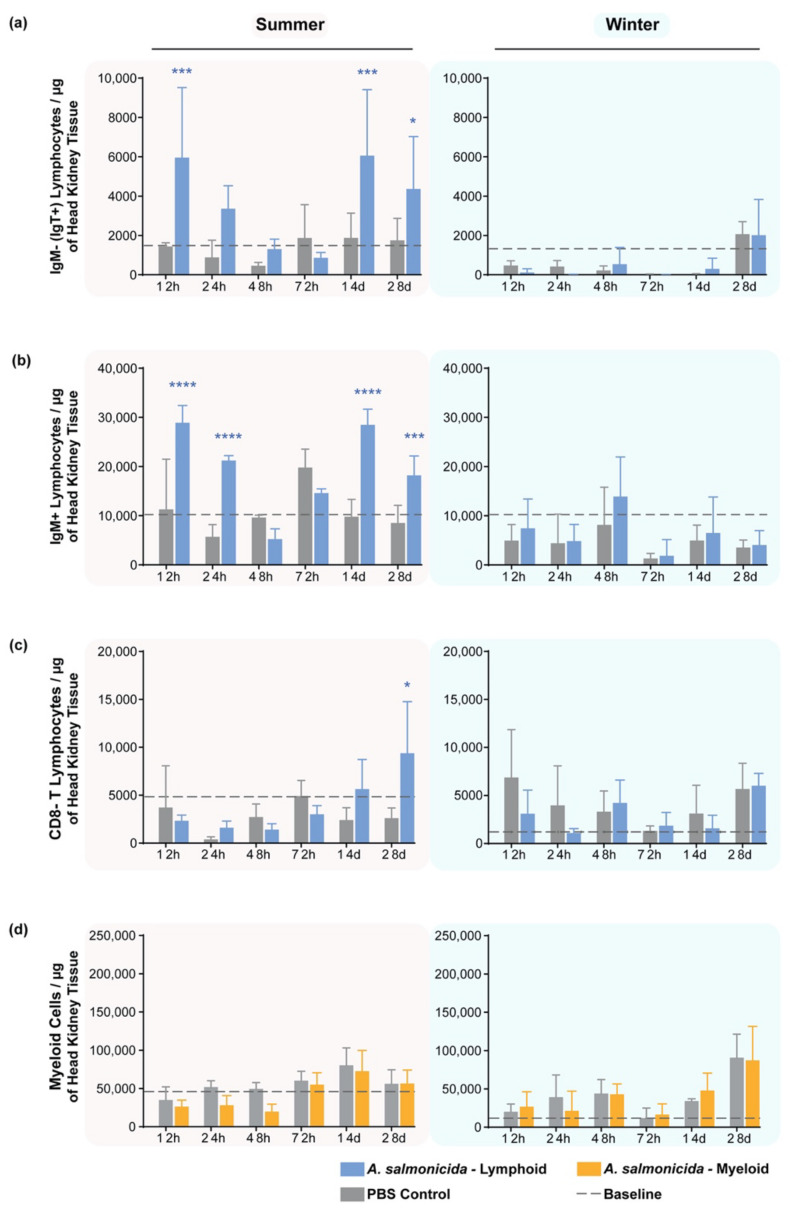
Cell response in the head kidney after stimulation with *A. salmonicida* analyzed by flow cytometry. (**a**) IgM^−^ lymphocytes/μg of tissue; (**b**) IgM^+^ lymphocytes/μg of tissue; (**c**) CD8^−^ lymphocytes/μg of tissue; (**d**) myeloid cells/μg of tissue measured in summer (**left column**) and in winter (**right column**). An increase in the number of IgM^−^ IgT^+^ B cells and IgM^+^ B cells was observed sporadically early (12 hps for both populations and also 24 hps for IgM^+^ B cells) and at 14 and 28 dps during summer. CD8^−^ T cells (“CD4”) were higher at 28 dps. No statistically significant trend was observed in the myelocytic response in both seasons. Baseline levels of the cells are from 5 naïve fish. Y-axes represent the number of cells adjusted to the µg of head kidney tissue collected. X-axes represent the days since injection of either the bacterium or PBS. Bars are color-coded according to whether they show the lymphoid (blue) or myeloid (yellow) cell populations after stimulation. The gray bars are data for the corresponding myeloid or lymphoid cells from fish exposed to PBS only. Statistical significance was calculated with a two-way ANOVA multiple comparisons test, with a Bonferroni post hoc correction test. *n* = 5 for each time point. * indicates *p* < 0.05; *** indicates *p* < 0.001; **** indicates *p* < 0.0001. The data are presented as mean values ± standard deviation (SD).

**Figure 6 biology-11-00174-f006:**
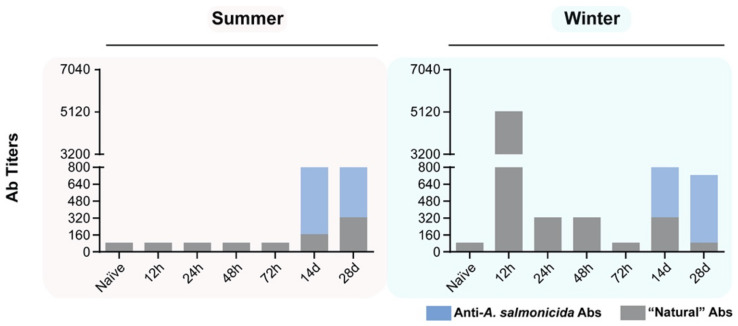
Antibody titers specific for *A. salmonicida* (in blue) compared to “natural” antibodies reactive against DNP-KLH (in gray) in sera from rainbow trout stimulated with *A.salmonicida*. Specific antibodies against *A. salmonicida* were detected in both seasons starting at 14 dps. A spike in “natural” antibodies was measured in winter at 12 hps; afterward, the “natural” antibody titer decreased over time. The y-axes represent antibody titer, whereas the x-axes represent the hours or days since animals were stimulated with *A. salmonicida*. The serum of 5 fish individually was used to measure the antibody titer for each time point.

## Data Availability

Not applicable.

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
