# Peer review of "Variations in Rainbow Trout Immune Responses against A. salmonicida: Evidence of an Internal Seasonal Clock in Oncorhynchus mykiss"

_biology, 2022, doi:10.3390/biology11020174_

Round 1
Reviewer 1 Report
General comments
In general, the subject is well presented, and the importance of conducting such research has been thoroughly explained. However, some parts of the Method need to be clarified
In the following section, I have some additional questions and remarks which I believe can improve the quality of this manuscript.
Major comments
Line 114: The authors mentioned the weight of trout ranged 20-100g, which is high heterogeneity. Could the author provide mean and standard deviation instead?
Line 114 -116: Rewrite this sentence
Line 116-118: More information about abiotic conditions should be mentioned as the authors experimented on the controlled condition. Some of them should be pH, dissolved oxygen, NH3 and NO2-. Were they maintained at a consistent level throughout the summer and winter trials?
Line 117: What does “a partially recirculating water system” mean?
Line 118-120: What was the extract duration for the summer and winter trials? As I understood, each took 120 days (June to September and January to March). I concern the duration of the trial, thus the size of rainbow trout at the end of each trial. Could fish size be an influential factor driven the change in innate immune?
Line 121: “A total of 130 fish were used for the experiments”. The authors should indicate details about fish size at the beginning and end of the trial.
Line 122-123: How did the authors sample these fish? What criteria for the selection of these fish? More details should be provided?
Line 185: what does this “()’ mean?
Line 200-202: This sentence is not necessary
Line 222: How about “**”? Was it P < 0.01? The author should provide statistical analysis in the Method specifically. For example, why did the authors use Mann-Whitney test? Was the data non-parametric? Did the author test the normal distribution of the data?
Line 253-254: Same comment as indicated in Line 222
Reviewer 2 Report
I found the manuscript really well written and interesting, presenting novelty data regarding rainbow trout seasonal immunity. In my opinion the paper is ready to publication after editing check
Reviewer 3 Report
This MS examines immune cell titers in rainbow trout challenged with the teleost pathogenic bacterium Aeromonas salmonicida, and attempts to tease apart this immune response from seasonal environmental variations by keeping experimental fish at a constant water temperature and light regime.
The work will be of interest to researchers studying seasonal fluctuations in the teleost immune system. For me, the most interesting result is that some teleost circannual rhythms are present even when holding the environmental conditions constant. The paper is well written, although a little long. I have challenges with some of the terms used, experimental design, statistical analysis and conclusions, as detailed below. Once these are addressed in a revised version, the MS will be suitable for publication in MDPI Biology.
Terms used: “Kinetics” is repeatedly inappropriately used as a synonym for “concentration” or “number”. To me, kinetics implies you are measuring a rate (e.g., of enzyme activity), not counting cells like here. I suggest changing this term throughout. E.g., the title of section 3.4 (line 267) could be “Response of leukocyte populations….” rather than “Kinetics of…”.
Experimental design: One experiment was done with 5 fish per treatment, giving an experimental repetition number of 1. This is thin for making some of the rather definitive conclusions about cause and effect in the discussion and conclusions (see below). Ideally, all the experiments would be repeated a second and third time. n=5 fish here but these are pseudoreplicates from a single experiment. The authors don’t have to do these experimental replicates, but the limited experimental data should cause them to moderate the certainty of their arguments in the discussion and results.
Statistical analysis: Good detail is given for the statistical tests used to distinguish experimental groups from controls, but some details of the results are missing. E.g., in figure 1-5 legends, what are the bars: +/- SD? SEM? If *=p,0.05, what do 2, 3, and 4 asterisks mean? What are figures a, b, c and d in each case? Figures 1 and 2 are too small to read easily. In fig. 6, were results determined for pooled sera from 5 individuals? Why not measure antibody titers in each sample separately so you could run statistical tests?
Conclusions: The discussion contains a lot of speculation about many complex organismal responses (e.g., augmented immune response) that are based on the authors’ measurements of a single cell type. Although I appreciated them demonstrating their thorough review of the literature, I think the discussion and conclusions make too many definitive conclusions that are not justified by the data. It would also be helpful to divide the discussion into subsections with titles to increase readability. E.g., line 458-9 “Temperature and photoperiod are key parameters influencing immune seasonal variations” would make a very nice such subsection title.
Other minor issues:
Line 30 and 54: briefly mention here that A. salmonicida is a bacterial pathogen that severely impacts the health of wild salmonid populations.
65: …measuring the expression of sets of genes…
72: …white and red blood cell counts, plasma…
381: …between winter and summer in O. mykiss [15].
520: The discussion of vaccinating fish comes out of nowhere. Would this be for lab and/or farmed salmonids against A. salmonicida? Is this practical?
529: …fish, this may not be possible…
Round 2
Reviewer 1 Report
The authors have well addressed my comments. Yet, I have two minor suggestions to make the content more explicit and comprehensible for readers, as follows:
Line 284-285. Please, provide mean ± SD for each of those parameters
Author Response
Thank you very much for the comment.
For Oxygen ( 11,5mg/l +/- 0,5) and pH ( 7.1 +/- 0.1) we do have the data. This could be added in the requested line. For nitrite and nitrate we used indicator paper which does not give exact numbers but a color which refers to physiological or unphysiological levels. But due to the daily change of about 1/3 of the water and the low numbers of fish there was no indication of problems with these two Parameter within the whole experimental period.